behaviour/ecology/systems biology

cooperative hunting, signalling, recruitment, lionfish

**Author for correspondence:**
Hanaa Sarhan
e-mail: hanaa.sarhan@unine.ch

# No evidence for conspecific recruitment for cooperative hunting in lionfish *Pterois miles*

## Hanaa Sarhan and Redouan Bshary

Institution of Biology, University of Neuchatel, Emile-Argand 11, Neuchatel 2000, Switzerland

HS, 0000-0002-0943-7052; RB, 0000-0001-7198-8472

Lionfish are common piscivores in the Indo-Pacific and invasive in the Caribbean. A fin flaring pattern, involving a rapid undulation of the caudal fin and sequential turning of both pectoral fins, was described in zebra lionfish as a signal to initiate cooperative hunting, and it was hypothesized that such hunting tactics may also exist in other lionfish species and contribute to their successful invasion in the Caribbean. Here, we investigated one of those invasive species, *Pterois miles*, in its natural range in the Red Sea. We did not observe evidence for cooperative hunting in the field. We complemented field observations with a laboratory experiment aimed at inducing subjects to recruit partners for cooperative hunts, exposing subjects to inaccessible prey in transparent housing as well as to a potential partner. We regularly observed the fin flaring pattern, but importantly, it was not directed at the partner. Thus, rather than being a signal, the fin flaring movement pattern seems to be a swimming mode in a confined environment. Furthermore, the two lionfish did not aggregate at the prey housing, reinforcing the field results that this species in the Red Sea does not depend on cooperation to hunt fish.

## 1. Introduction

One way in which animals may benefit from grouping is because of the transfer of information regarding the distribution of patchy food [1]. The transfer of information may be passive, i.e. based on cues, or active due to communication. Active communication warrants shared interest between signaller and receiver. One possible cause is kin selection, like in the waggle dance in honeybees [2] or pioneer ants leaving trails of pheromones that lead foraging ants between food-rich areas and their nests [3]. Alternatively, recruiting partners yields direct benefits to the signaller. For example, ravens recruit conspecifics to overcome defence of food by dominant individuals and competitors [4].

Recruitment may also occur between individuals belonging to different species. Groupers may perform a headstand signal to attract various partner species (moray eels, Napoleon wrasse or octopus) to a location where prey is hiding in a crevice, eliciting a cooperative hunting attempt on that prey [5]. Cooperative hunting occurs when two or more individuals interact in pursuing a hunt that eventually produces on average greater payoffs per individual than solitary hunting [6]. Correlative evidence for a positive relationship between hunting group size(s) exists for a variety of mammals and birds [7].

Much research on cooperative hunting has focused on the complexity of the coordination between hunters. Coordination can range from spontaneous 'simultaneous single hunts' to individuals planning a hunt (intentional hunting) in which they play different roles (termed collaboration) [7,8]. Intentional and collaborative hunting was initially reported in few endotherm vertebrate species (Harris hawks [9]; chimpanzees [8]; lions [10]; dolphins [11]) and hence assumed to rely on advanced cognitive abilities that require at least an endotherm vertebrate brain [8]. However, the more recent description of intentional communication and collaborative hunting in various fish species [5,12,13] changed this view. A case study yielding seemingly particularly complex hunting patterns in a fish described communicative and cooperative hunting in zebra lionfish (*Dendrochirus zebra*) [12]. Building on field observations that lionfish species not only hunt alone [14] but also in the presence of conspecifics [15] and with moray eels [16], zebra lionfish were subjected to laboratory experiments [12]. In these experiments, subjects were initially exposed to inaccessible prey. The subjects were described to recruit naive conspecifics and heterospecific lionfish (*Pterois antennata*), confined behind an opaque barrier, to join the hunt. To do so, subjects approached the potential partner head on and pectoral fins flared, rapidly undulated their caudal fin with a slow and separate waving of the pectorals ([12], also see figure 1*c*). Subjects would then swim back to the prey location ('leading' behaviour) and return to a non-following partner to repeat the display [12]. Once the partner was released by the experimenter, the resulting pair would then hunt in highly coordinated ways on now accessible shoaling prey, alternating attacks in what appeared to be a reciprocal manner [12]. The findings were interpreted as evidence that zebra lionfish display highly sophisticated social behaviour to achieve high hunting success rates [17]. Furthermore, as recruited *P. antennata* behaved very similarly, the authors also hypothesized that sophisticated cooperative hunting may explain the high feeding rates of *Pterois* lionfish species in their invasive range in the Caribbean.

*Pterois miles* and *P. volitans* were introduced in Florida water in the 1980s, apparently from home aquaria [18]. The following invasion across the Caribbean has attracted great attention by conservationists as the negative effects are very strong [19]. Some parameters have already diverged from the native populations in the Indo-Pacific: lionfish in the native range exhibit slower life histories and/or narrower resource use [14–16]. One proposed reason for lionfish being successful hunters in their invasive range is that they display a rather unique combination of adaptations to hunting that Caribbean prey had not encountered before. Lionfish show striped coloration for crypsis [20], slow movement to avoid prey flight responses [21–24] and they blow water towards prey to make it orient against the resulting current and hence towards its predator [20]. The hypothesized cooperative hunting with active recruitment of partners based on the study of zebra lionfish [12,15] would hence add to an already fearsome list of features.

Given that *P. miles* is one of the two lionfish species of Indo-pacific origin that is found in the Caribbean, we decided to investigate whether this species exhibits communicative cooperative hunting in its native range that is similar to what has been described for zebra lionfish. Only after our plans had been made, a note of concern was published indicating that the data by Lönnstedt *et al.* [12] may be at least in part manipulated and/or fabricated, as the number of subjects is higher than the records for captured fish [25,26]. We conducted a series of field observations and laboratory experiments. First, we investigated the frequency and duration of any *P. miles* interspecific hunting associations in the field, by use of snorkelling and diving. Second, we explored whether the fin flaring pattern described for the Australian zebra lionfish *D. zebra* is also shown by *P. miles* in the Red Sea, using both field observations and a laboratory experiment (see electronic supplementary material, S1 videos on *P. miles* and video by Lönnstedt *et al.* [12] for *D. zebra*). The experiment was based on the design by Lönnstedt *et al.* [12], aimed at testing whether *P. miles* actively recruit naive partners to hunt in pairs. We manipulated prey availability (present or absent) and partner availability (present initially in a separate compartment, or absent) and recorded the occurrence of fin flaring patterns. If the fin flaring patterns serve as a recruitment signal, we expected a significant interaction where fin flaring should be frequent near the separate compartment if both prey and a partner are present.

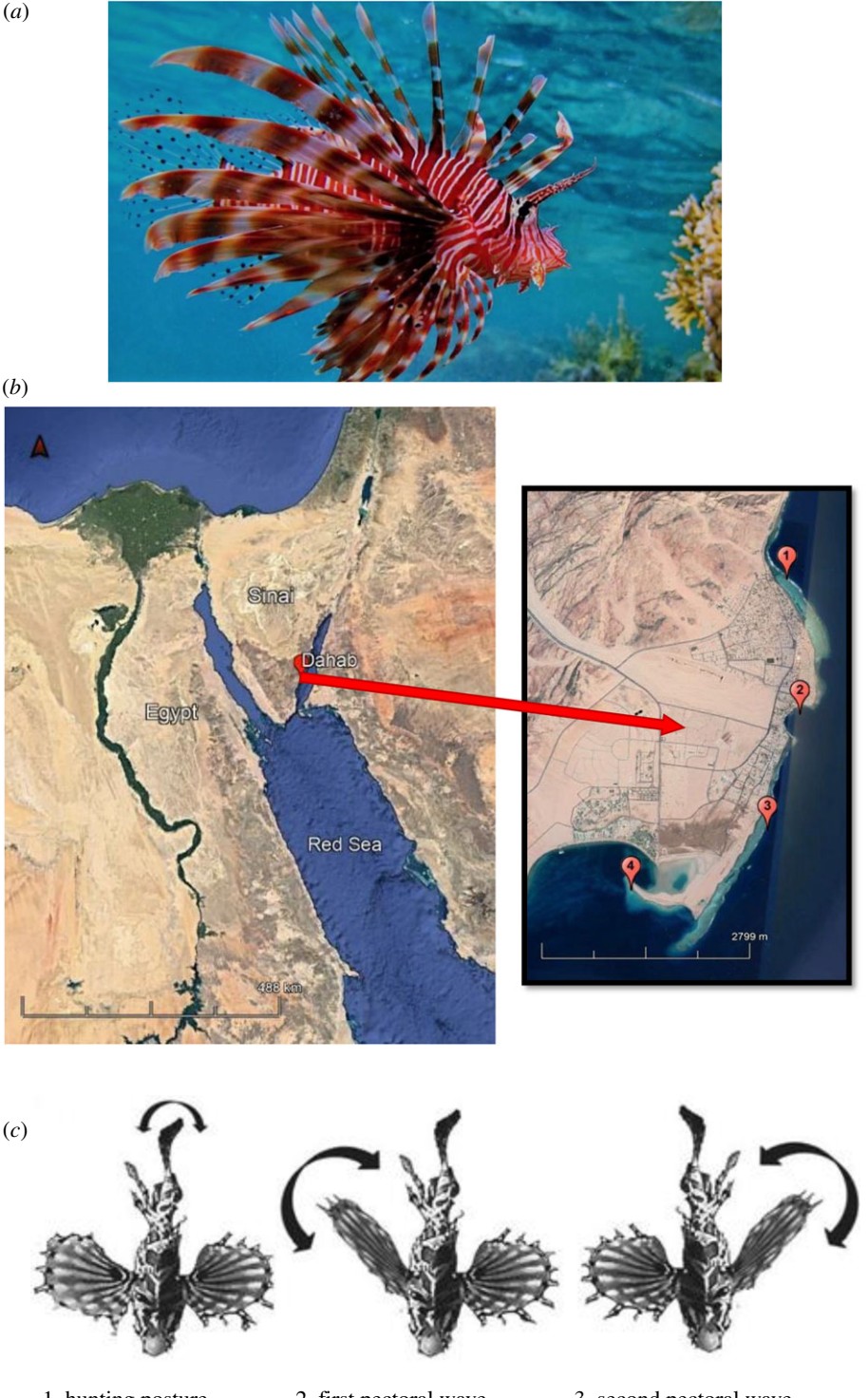

**Figure 1.** (*a*) The study species *P. miles* (Photo credit: H.S. Three pinnacles, Dahab, Egypt). (*b*) Location of four reef sites surveyed for lionfish species fin flaring behaviour in the northern Red Sea region. Image modified from Google earth v. 6.2 (2021). (*c*) The flared fin display sequence by initiator predators: where one individual approaches partner head on, it rapidly undulates its caudal fin for 3–9 s then followed by a slow and separate waving of the flared pectorals (Photo credit: Lönnstedt *et al.* [12]).

## 2. Material and methods

### 2.1. Field observations

Lionfish *P. miles* (figure 1*a*) were observed on four reefs near Dahab city located in the northern Red Sea (figure 1*b*), from August to November 2017. The four sites for data collection were (1) Assalah coast with

estimated area of 0.09 km$^2$, (2) light house (0.06 km$^2$), (3) Soliman reef (0.09 km$^2$), and (4) three pinnacles (0.06 km$^2$), all located in Dahab city, South Sinai, Red Sea, Egypt (electronic supplementary material, figure S1A). Field observations were conducted via snorkelling and diving during the crepuscular times of the day (4.30–6.00 and 18.00–19.30), i.e. when the species is most active [24,25]. Only active individuals were picked in an opportunistic way, i.e. the first individual found was followed for a maximum of 30 min, and then the next encountered active individual was followed. A distance of 3 m was respected to avoid disturbance. The length of focal individuals as well as associated individuals were estimated in centimetres with the help of a reference scale fixed on the observer's fin. During observations, lionfish swam in depth ranging from 1 to 15 m.

Regarding the maximum of four follows during a single observation session, we are confident that different individuals were sampled. By contrast, the same individual may have been resampled during repeated visits to the same locations, as we did not recognize lionfish individually. The lionfish density in the Dahab area is very high, with an estimated 87.3 ± 117.3 individuals per 0.01 km$^2$ [27]. Therefore, the estimated number of lionfish at each site was 785, 524, 785 and 524, respectively, and our 302 individual follows comprised 72, 74, 78 and 78 follows at the four sites, respectively. Based on these numbers, we estimated that repeated sampling of individuals was probably rare. We are, therefore, confident that our results represent what lionfish typically do at our study sites, despite the small amount of uncontrolled pseudoreplication.

The most basic measure was to quantify how much time lionfish spent in association with conspecifics and/or record any occasions of associations with moray eels, as these had been described by Naumann & Wild [16]. We used 1 m maximal distance as an association criterion [28,29] and asked whether lionfish spend more time in such distance to conspecifics than expected by chance as an indicator whether or not they seek each other. We looked for the fin flaring pattern described by Lönnstedt et al. [12] during the entire observation period. Similarly, we recorded any attempt to herd prey by spreading the pectoral fins like a shield [16], as this spreading almost always preceded any foraging attempt. Such attempts sometimes included orienting the head towards a nearby prey (typically close or on a substrate, i.e. reef or the sandy bottom), mostly accompanied with a rapid mouth opening to suck prey in. Note, however, that it was impossible to ascertain whether or not a fish was caught in these events.

## 2.2. Laboratory experiment and fish collection

The laboratory experiment was conducted at Open Ocean Science Center (OOSC) at Dahab city in September to November 2019. Fish capturing, handling and acclimatizing process was performed carefully to minimize stress responses and to provide a neutral environment for the captured individuals. Forty-eight lionfish individuals were captured using handmade plastic nets and gloves from depths from 0 to 10 m. Individuals were brought slowly to the surface to avoid air bladder inflation. Fish were directly transferred into holding tanks for 3 days without being fed, then assigned into the experimental tanks. The round experimental tanks were 153 cm in diameter and 60 cm high, with the water level at 40 cm (figure 2a). A semi-closed system was used to maintain continuous seawater change with a flow rate of 2 l min$^{-1}$, and an air pump as well as a water pump to provide oxygen and some current.

In each tank, a pair of lionfish was placed for 10 days to acclimatize before the experimental trials. Starting from the third night, individuals were fed twice during the crepuscular time two pieces (0.5 × 0.5 cm) of dead shrimps for each individual, while during experimental days, individuals were fed only after trials to ensure that they were motivated to hunt during trials. Aquaria were cleaned once a day to prevent biotic accumulation. All fish were released at sites of capture after completing the experiment.

In parallel with collecting lionfish individuals, we caught 25 blue–green chromis (Chromis viridis), using a barrier net 1.5 × 1.5 m with a 0.5 × 0.5 cm mesh size to present them as potential prey to lionfish. The chromis were kept in a single aquarium (50 × 30 × 30 cm), and left to acclimatize for 7–10 days. The aquarium was provided with seawater open system flow (2 l min$^{-1}$), and an air pump to provide appropriate aeration. An artificial coral block and small PVC tubes were added to serve as shelters and blue background was placed all around the aquarium. Fish were fed twice per day with commercial prawn flakes, food residuals were removed daily, and tank surfaces were cleaned at least every third day.

(a)

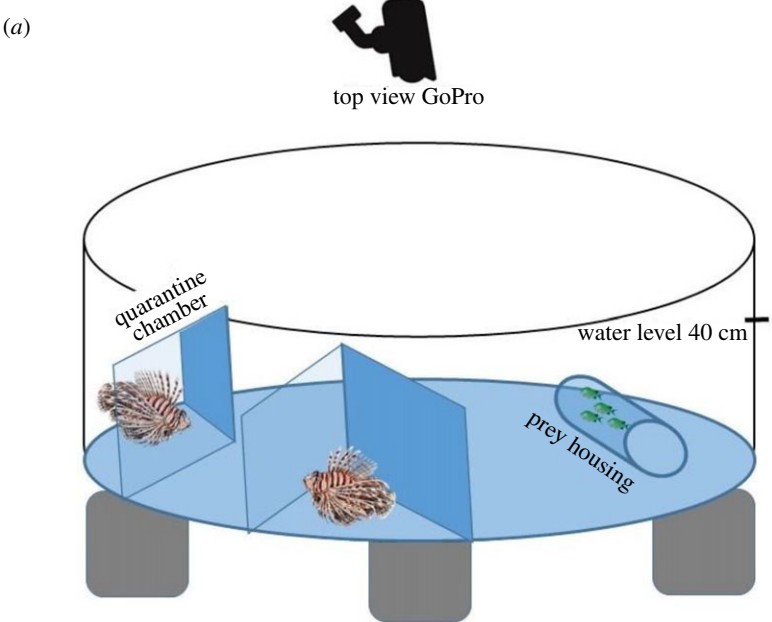

(b)

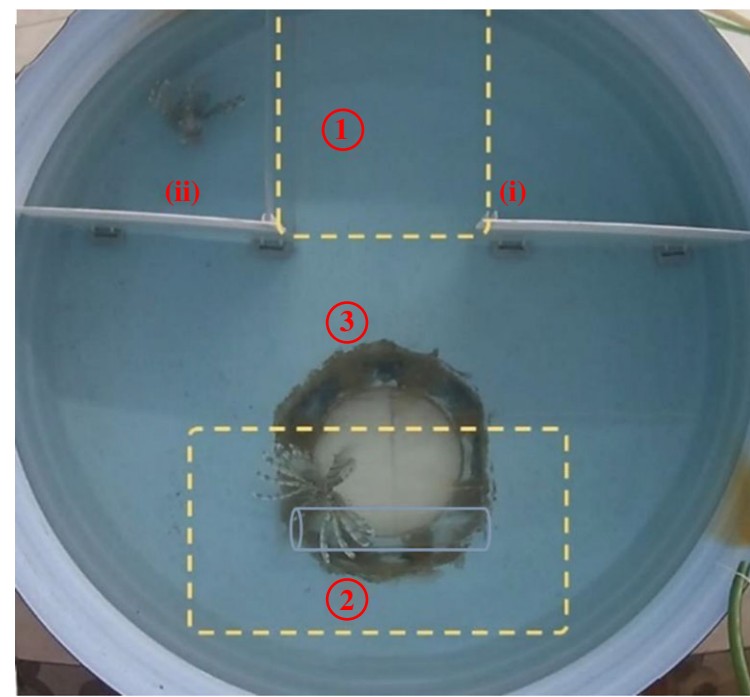

**Figure 2.** Tank set-up to test recruitment tendency of lionfish. (a) An illustration of the initial tank structure prior to the experiment. (b) A life shot of the zone references after focal fish were released, and the quarantined partner has no information about the existence of prey, where (i) focal chamber and (ii) partner chamber, and zones (1) partner zone, (2) hunting zone—the transparent prey housing is edited with grey border for reader's visibility—and (3) neutral zone. The round circle located in the hunting zone is dedicated to further studies.

### 2.2.1. Experimental design

The design of the experiment and the set-up of the round experimental tanks were inspired by the study of Lönnstedt *et al*. [12], the main difference being that we had round tanks. Inside each tank, we placed two holding compartments separated by a channel (figure 2a,b). Pairs of individuals were locked separately inside the compartments with the ability to see each other through two removable transparent barriers.

By contrast, the barrier facing the rest of the tank was opaque, preventing locked individuals from seeing the location where a tube containing either four prey fish (experimental treatment) or nothing (control treatment) would be presented. Consequently, a released focal individual trying to catch a prey would not be able to see the potential partner locked away behind the barrier through the opaque barriers installed facing the prey housing. The tube was placed close to the opposite side of the holding compartments, such that lionfish could still swim all around it (figure 2b).

We ran experiments in a full 3 × 2 design, yielding six conditions. The focal individual could be in three association states regarding its partner, referred to as 'social state' from now on: (i) singleton: the partner had been removed to another tank (100 × 60 cm) before the trial; (ii) partner locked: the partner was contained in a holding compartment; and (iii) partner released: the partner could swim around freely in the tank. The second and third conditions were tested within the same trial, invariably starting with the partner locked. Regarding our second-factor prey availability, the Plexiglas tube either contained four prey fish (prey present) or no prey fish at all (prey absent). In order to be able to define a visit, as well as to define hunting activity by either the focal or its partner, we divided the tank into three zones (figure 2b): the 'partner visit zone' was the corridor between the two holding compartments. The 'hunting zone' was a rectangle 60 × 100 cm around the lying Plexiglas tube. The 'neutral zone' as a general activity zone contained all the other areas of the tank.

### 2.2.2. Experiment protocol and data collection

All trials took place at night as lionfish were more active during that time compared with during the day. Nevertheless, white light wrapped with towels was put on to allow direct observation as well as filming. A GoPro Hero +3 camera was installed 130 cm above the rim of the tank to film the trials. Before each trial, the focal individual and its partner (if present) were guided into the holding compartments, the compartments being randomly assigned. This was achieved by gently pushing objects (a Plexiglas sheet or stick) behind them. Lionfish were then left to settle for 10 min. Meanwhile, we placed a Plexiglas tube (40 × 15 cm) in the tank at its predetermined location. The tube had a removable cap to allow the insertion of prey for trials in which prey was presented to the lionfish. The four prey fish were always randomly caught from the holding tank; individual identity was not known. A trial started by removing the transparent barrier of the focal's holding compartment. Focal individuals left the compartment within the next 60 s. In the singleton condition, a trial lasted 15 min. With a partner, the focal invariably started to hunt alone. If it revisited its partner, scored as entering the channel between the two holding compartments, the partner would be released 5 min later. This delay in the release gave us opportunity to record any repeated signalling if it occurred. If the focal individual did not visit the partner, the partner would be released after 15 min. The total length of a trial with partner was 20 min, so the two conditions 'partner locked' and 'partner released' occurred with variable durations. The focal and partner compartments were considered as part of the neutral zone once the partner had been released. A trial ended with the removal of the Plexiglas tube; and any prey was returned to their holding tank.

The following information was extracted from the videos, using Boris 7.9.7 software: (i) any fin movement sequence as described as fin flaring signal by Lönnstedt et al. [12] (figure 1b) and its context (which zone and location of partner), and (ii) the time from releasing the focal to its returning to the 'partner visit zone'. See electronic supplementary material for statistical analysis.

## 2.3. Statistical analysis

### 2.3.1. Field observations

We used Waser's gas model (WGM) [28–30] as the simplest option to calculate a null hypothesis regarding the mean duration of associations between lionfish. Using the model to calculate the predicted mean duration needs rather minimal information, namely the average swimming speed ($v$) of individuals, and a set maximal distance criterion ($r$) between two individuals that they are still considered to be associated. Application of the model warrants to consider movements only in a two-dimensional space, which works well for an observer watching from above. Under these assumptions, the mean duration of associations ($t$) predicted if individuals move independently of each other is given by the below equation

$$t = \frac{1.745\,r}{v}. \tag{2.1}$$

We used a mean speed of 3 cm s$^{-1}$, as reported by Dahl & Patterson [31], and 100 cm as the distance criterion [28,29].

### 2.3.2. Laboratory experiments

In general, all our data represent independent measures as each fish was tested only once. As we had several data points per fish in the test, individual ID was added as a random factor. Data distributions were negatively binomial and the normality was always violated. As our data contained zeros, we applied generalized linear mixed-effect models using Template Model Builder with the β zero-inflation properties to calculate the frequencies, time and proportions (GLMM: 'glmmTMB'). The data and the R codes as well as a videos of fin flaring and recruitment tendency are available to the public on the Dryad Digital Repository at https://doi.org/10.5061/dryad.w3r2280q8.

The frequency of the fin flaring was analysed as a function of prey presence/absence, social state (singleton, partner locked and partner released) and the focal individual's location (hunting zone, neutral zone and partner zone). The focal and partner ID as well as observation time were random factors. If the fin flaring movement pattern serves as a signal to initiate cooperative hunting, we expected a three-way interaction. If lionfish try to recruit the partner to guide it towards prey, the pattern should occur most often when prey is present, in the partner zone with a partner locked up. Alternatively, if lionfish try to make partners join at the prey location, the pattern should occur most often when prey is present, in the hunting zone with a partner locked up. For the analyses, we quantified how much time the focal spent in each zone in each condition. We then used the occurrences of fin flaring in each zone and time spent in each zone to calculate fin flaring frequencies.

As an additional test whether lionfish are likely to seek a partner when confronted with inaccessible prey, we analysed whether the time from release to returning for the first time to the 'partner zone' varied as a function of prey presence/absence and partner absent/locked. We obtained a single value in seconds for each trial. If lionfish seek the presence of a partner for joint hunting, we expected that subjects will revisit the partner zone soonest if there is prey and a partner locked up. Subject and partner IDs were added as a random variable.

## 3. Results

### 3.1. Field observations

We collected a total of 302 follows, yielding 172 h of observations. In general, lionfish spend 14.3% of time in association with conspecifics. The mean duration of the associations was 4.2 ± 1.4 min (figure 3). Based on the gas model, assuming an average speed of 3 cm s$^{-1}$, the expected mean duration of an association was 59 s. There is thus some indication that associations last approximately four times longer than predicted by chance. However, the more critical question was whether lionfish would hunt together when in association. We observed a total of 103 events of lionfish spreading their fins like a shield. In 12 of these 103 cases, a conspecific was within 1 m, i.e. slightly less than the 14.7 cases expected by chance if associations do not serve to hunt cooperatively. No fin flaring sequences were recorded during any association, and the sequence was only observed once in a singleton.

### 3.2. Laboratory experiments

Results were based on a total of 48 individuals that formed 24 pairs. We observed a total of 1512 fin flaring, yielding 48 h of observations. We regularly observed the fin flaring pattern during trials, if it serves as a recruitment signal, we expected it to occur most frequently in the partner zone but only if there is prey and the partner locked up. However, we did not find such a three-way interaction but instead two two-way interactions (figure 4a). As expected if fin flaring was a recruitment signal, it occurred most frequently in the corridor between the two holding compartments when prey was present (two-way interaction, $\chi^2 = 6.49$, $p = 0.039$). However, opposite to the recruitment hypothesis, fin flaring occurred most frequently in the corridor after the partner had been released and hence was not in the zone any more (two-way interaction, $\chi^2 = 12.2$, $p = 0.016$).

Regarding the time focals spent in the three zones, we found a significant three-way interaction opposite to what would be expected if lionfish hunt cooperatively: focals spent more time in the hunting zone (and less time in the neutral zone) if there was prey and the partner was still locked up

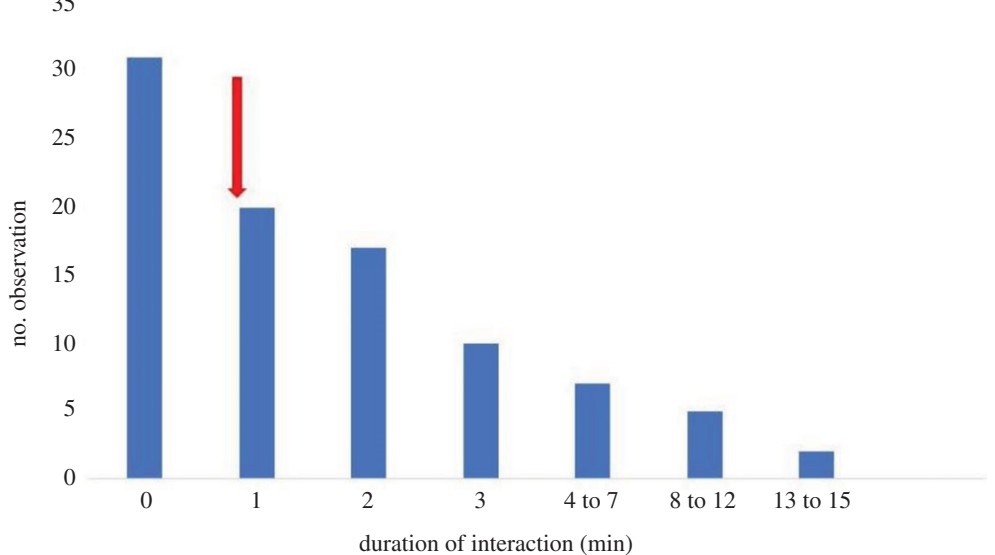

**Figure 3.** Observed frequency distributions of durations (min) of associations between lionfish conspecifics. The *x*-axis shows different time categories that were grouped in a nonlinear fashion. The arrow almost above the 1 min time category indicates the average duration of associations (59 s) predicted by a null model, assuming independent movements of individuals observed.

($\chi^2 = 10.08$, $p = 0.039$). Focals spent little time in the hunting zone when the partner was released, irrespective of prey presence/absence (figure 2*b*).

The time it took focals to return to the partner zone was explained by an interaction between prey condition and social state ($\chi^2 = 10.06$, $p = 0.0015$). Focals took longest to return if there was a partner locked up and prey absent (figure 2*c*). In the presence of prey, return times were similar between partner locked and the singleton condition.

## 4. Discussion

We had asked whether lionfish *P. miles* in the Red Sea show any evidence for cooperative hunting based on active recruitment, as such hunting techniques might have helped them to become such a highly successful invasive species in the Caribbean. While the fin flaring pattern described in *D. zebra* was virtually absent in *P. miles* in the field, it occurred frequently during our laboratory experiments. The field data contrast with Lönnstedt *et al.* [12] who apparently observed the pattern regularly in nature (though no data were presented). The two key differences between the two studies and hence the two species were that we found no evidence that the fin flaring is used as a signal and that *P. miles* individuals would coordinate to hunt cooperatively. A hypothesis for the frequent occurrence of the fin flaring pattern in *P. miles* that needs further investigation is that its occurrence is linked to manoeuvering [32–35]. The main current argument in favour of this hypothesis is that it occurred mostly in the narrow partner zone in our experiment. Hence, the question arises whether the fin flaring pattern could also be used for manoeuvering in *D. zebra*. Speaking against this hypothesis are the observations that partners apparently responded to a visit and fin flaring pattern with their own one (see figure 2*b* in [12]). On the other hand, the interaction site between the test fish and its contained partner had to happen in a narrow place, like in our experiment. Furthermore, we are uncertain how to interpret the information provided by Lönnstedt *et al.* [12] in the electronic supplementary material, video that shows the fin flaring pattern. We invite colleagues to watch it for themselves (electronic supplementary material, video S1A): in our view, it looks like the focal individual used the pattern to orient towards a prey that is subsequently eaten, an interpretation that favours the manoeuvering hypothesis.

The space use of *P. miles* subjects during experiments conforms with the interpretation that this species does not hunt cooperatively, at least not at our study site in the Red Sea. Our *Chromis* prey can be considered attractive, just as *Apogon* sp. is for *D. zebra* [36–38], and both were presented in small groups. Prey was behind a transparent barrier in the study by Lönnstedt *et al.* [12], while inside a transparent tube that could be approached from all sides in our study. Though the details of the

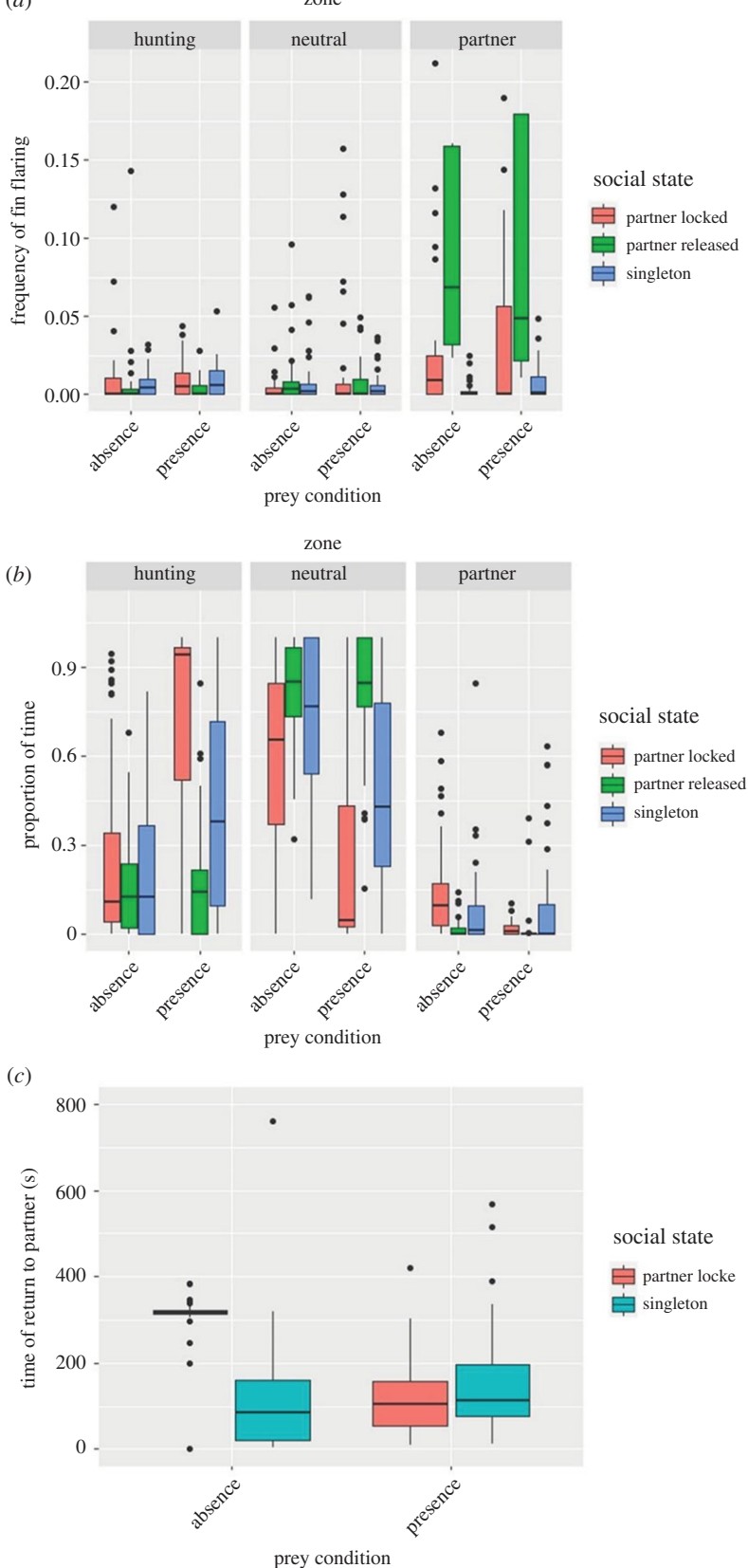

**Figure 4.** Experimental results. (*a*) The frequency of fin flaring patterns and (*b*) proportional space use; both variables as a function of location (hunting, neutral and partner zones) and absence/presence of prey. (*c*) The time in seconds it took subjects from being released to return to the partner zone, depending on whether prey was absent or present and whether a partner was locked up or the subject was singleton.

presentation hence differed, subjects in both studies experienced that prey is inaccessible when hunting alone. Thus, despite the fact that various variables were slightly different (prey species, aquarium size, transparent housing structure), we think that the situation should have triggered partner recruitment in our study as well. However, in our study, inaccessible prey and a locked-away partner did not result in a quick return to the partner (electronic supplementary material, video S1B), which contradicts the hypothesis that *P. miles* recruits partners for joint hunting proposed in Lönnstedt study.

In line with the lack of evidence for recruitment, we did not find any evidence that *P. miles* hunts cooperatively in nature. While associations in the field lasted on average longer than expected by chance, they were much shorter than, for example, between collaboratively hunting fish, like groupers and moray eels [39] or yellow goatfish [40,41]. Also, during experiments, the lionfish did not aggregate in the hunting zone, which suggests a rather chance character of lionfish associations [14,42]. Taken together, our results suggest that the success of invasive *P. miles* in the Caribbean is not based on cooperative hunting but on other features. Clearly, the reported variation in cooperative hunting between lionfish species and/or study sites warrants further investigation.

Ethics. Ethics and research permit was granted by Ras Mohamed National Park office before starting the study.
Data accessibility. Data available from the Dryad Digital Repository: https://doi.org/10.5061/dryad.w3r2280q8.
The data are provided in electronic supplementary material [43].
Authors' contributions. Both authors designed the study. H.S. collected all data and wrote a first draft, which was finalized by R.B.
Competing interests. We declare we have no competing interests.
Funding. Research was funded by a Swiss Government Excellence Scholarship to H.S. and a Swiss National Science foundation grant to R.B. R.B. is supported by the Swiss Science Foundation grant 310030_192673/1.
Acknowledgements. We thank Radu Slobodeanu for statistic support, the Open Ocean science station in Dahab for logistic support and Ahmed Hassan for fish catching support.

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
