## [Peer Review File · Royal Society Open Science]

Review History

RSOS-210828.R0 (Original submission)

Review form: Reviewer 1

Is the manuscript scientifically sound in its present form?

Yes

Are the interpretations and conclusions justified by the results?

Yes

Is the language acceptable?

Yes

Do you have any ethical concerns with this paper?

No

Have you any concerns about statistical analyses in this paper?

No

Recommendation?

Accept with minor revision (please list in comments)

Comments to the Author(s)

I reviewed this manuscript upon its first submission to Biology Letters. Before I comment on this version of the manuscript, I first wanted to comment on some of the initial reviewer comments, and subsequent rejection of the manuscript from Biology Letters. One of the reviewers on the initial submission commented that the manuscript contained negative results and would be more appropriate for a specialized journal, despite the fact that this manuscript was a direct attempt to replicate the cooperative hunting behaviours reported in a previous Biology Letters paper that required a subsequent Expression of Concern. One of the major problems with science is that remarkable findings are hastily published with scant evidence, yet more robust studies are hastily rejected if they find less-glamorous results. This manuscript should have been published in Biology Letters to give it the same platform as the (problematic) lionfish cooperative hunting paper by Lonnstedt et al. (<https://www.sciencemag.org/news/2019/09/can-you-spot-duplicates-critics-say-these-photos-lionfish-point-fraud>).

Following from the above, I strongly disagree with the authors removing the section about the Lonnstedt et al. issue in the introduction of the manuscript. This is very important information that underscores why the present study was conducted, so the information should be put back into the introduction.

Aside from that major comment, I only have a few minor comments in addition to my last review.

Line 166 – fish were brought to the surface and placed immediately into holding tanks? Were they transported to a field lab first?

Line 383 – what is the transparent holding structure and why is that different from the Lonnstedt study?

I commend the authors for supplying data and videos on dryad. Please make sure the data are the raw data and they are organised well, with a metadata tab to explain each column. Please also make sure that all videos are available if anybody wants them, not just ‘representative’ videos of particular behaviours.

A thorough proofread is required to pick up several typos throughout.

Review form: Reviewer 2**Is the manuscript scientifically sound in its present form?**

Yes

Are the interpretations and conclusions justified by the results?

No

Is the language acceptable?

Yes

Do you have any ethical concerns with this paper?

No

Have you any concerns about statistical analyses in this paper?

Yes

Recommendation?

Accept with minor revision (please list in comments)

Comments to the Author(s)

I enjoyed reading this interesting study which carried out some field and lab work which gave no support for cooperative hunting in Pterois miles. Lionfish have attracted attention as an invasive species and the potential for cooperative hunting behaviour in this context is intriguing. I don't see the negative results of this study as a problem but would like to recommend more careful phrasing of some sentences (including the title).

I don't think the authors can conclude that "lionfish Pterois miles do not recruit conspecifics for cooperative hunting" because they only tested for a limited number of possibilities. A simple solution would be to change the title to "No evidence for conspecific recruitment for cooperative hunting in lionfish Pterois miles". This change of title would probably be the most appropriate way of dealing with a negative result.

In the Abstract the authors state "...reinforcing the field results that this species in the Red sea does not depend on cooperation to catch fish". As far as I could tell, the authors did not observe any lionfish capturing prey in the field. They only recorded association patterns and fin flaring.

Statistical analysis

I am puzzled by Waser's gas model as an analysis tool. How can this possibly provide a prediction for biological processes? What if the lionfish like particular spots and like most animals don't use the entire available habitat to the same extent? I don't see how Waser's model accounts for micro-habitat preferences and how this can be distinguished from active association. The information given on Lines 147-149 "The most basic measure was to quantify how much time lionfish spent in association with conspecifics and/or moray eels as described by Naumann and Wild [17]." is not very helpful because that is a single-page paper with no detailed description.

Minor points

Line 12, "Lionfish are highly successful piscivores"

Did you mean that they are highly piscivorous or that they are really particularly successful compared to other species? If so, it would be interesting to know which species are used for comparison.

Lines 32-33 "Animals may benefit from grouping because of the transfer of information regarding the distribution of patchy food [1]."

There are many reasons why animals can benefit from grouping. Rephrase "One way in which animals may benefit from grouping is because of the transfer of information regarding the distribution of patchy food [1]" or cover more of the benefits of group-living.

Decision letter (RSOS-210828.R0)

Dear Dr Sarhan

On behalf of the Editors, we are pleased to inform you that your Manuscript RSOS-210828 "Lionfish *Pterois miles* do not recruit conspecifics for cooperative hunting" has been accepted for publication in Royal Society Open Science subject to minor revision in accordance with the referees' reports. Please find the referees' comments along with any feedback from the Editors below my signature.

Please submit your revised manuscript and required files (see below) no later than 7 days from today's (ie 16-Aug-2021) date. Note: the ScholarOne system will 'lock' if submission of the revision is attempted 7 or more days after the deadline. If you do not think you will be able to meet this deadline please contact the editorial office immediately.

on behalf of Professor Kevin Padian (Subject Editor)
openscience@royalsociety.org

Reviewer comments to Author:

Reviewer: 1

Comments to the Author(s)

I reviewed this manuscript upon its first submission to Biology Letters. Before I comment on this version of the manuscript, I first wanted to comment on some of the initial reviewer comments, and subsequent rejection of the manuscript from Biology Letters. One of the reviewers on the initial submission commented that the manuscript contained negative results and would be more appropriate for a specialized journal, despite the fact that this manuscript was a direct attempt to replicate the cooperative hunting behaviours reported in a previous Biology Letters paper that required a subsequent Expression of Concern. One of the major problems with science is that remarkable findings are hastily published with scant evidence, yet more robust studies are hastily rejected if they find less-glamorous results. This manuscript should have been published in Biology Letters to give it the same platform as the (problematic) lionfish cooperative hunting paper by Lonnstedt et al. (<https://www.sciencemag.org/news/2019/09/can-you-spot-duplicates-critics-say-these-photos-lionfish-point-fraud>).

Following from the above, I strongly disagree with the authors removing the section about the Lonnstedt et al. issue in the introduction of the manuscript. This is very important information that underscores why the present study was conducted, so the information should be put back into the introduction.

Aside from that major comment, I only have a few minor comments in addition to my last review.

Line 166 – fish were brought to the surface and placed immediately into holding tanks? Were they transported to a field lab first?

Line 383 – what is the transparent holding structure and why is that different from the Lonnstedt study?

I commend the authors for supplying data and videos on dryad. Please make sure the data are the raw data and they are organised well, with a metadata tab to explain each column. Please also make sure that all videos are available if anybody wants them, not just ‘representative’ videos of particular behaviours.

A thorough proofread is required to pick up several typos throughout.

Reviewer: 2

Comments to the Author(s)

I enjoyed reading this interesting study which carried out some field and lab work which gave no support for cooperative hunting in Pterois miles. Lionfish have attracted attention as an invasive species and the potential for cooperative hunting behaviour in this context is intriguing. I don't see the negative results of this study as a problem but would like to recommend more careful phrasing of some sentences (including the title).

I don't think the authors can conclude that “lionfish Pterois miles do not recruit conspecifics for cooperative hunting” because they only tested for a limited number of possibilities. A simple solution would be to change the title to “No evidence for conspecific recruitment for cooperative hunting in lionfish Pterois miles”. This change of title would probably be the most appropriate way of dealing with a negative result.

In the Abstract the authors state “...reinforcing the field results that this species in the Red sea does not depend on cooperation to catch fish”. As far as I could tell, the authors did not observe any lionfish capturing prey in the field. They only recorded association patterns and fin flaring.

Statistical analysis

I am puzzled by Waser's gas model as an analysis tool. How can this possibly provide a prediction for biological processes? What if the lionfish like particular spots and like most animals don't use the entire available habitat to the same extent? I don't see how Waser's model accounts for micro-habitat preferences and how this can be distinguished from active association. The information given on Lines 147-149 “The most basic measure was to quantify how much time lionfish spent in association with conspecifics and/or moray eels as described by Naumann and Wild [17].” is not very helpful because that is a single-page paper with no detailed description.

Minor points

Line 12, “Lionfish are highly successful piscivores”

Did you mean that they are highly piscivorous or that they are really particularly successful compared to other species? If so, it would be interesting to know which species are used for comparison.

Lines 32-33 “Animals may benefit from grouping because of the transfer of information regarding the distribution of patchy food [1].”

There are many reasons why animals can benefit from grouping. Rephrase “One way in which animals may benefit from grouping is because of the transfer of information regarding the distribution of patchy food [1]” or cover more of the benefits of group-living.

===PREPARING YOUR MANUSCRIPT===

===PREPARING YOUR REVISION IN SCHOLARONE===

Author's Response to Decision Letter for (RSOS-210828.R0)

See Appendix A.

Decision letter (RSOS-210828.R1)

Dear Dr Sarhan,

I am pleased to inform you that your manuscript entitled "No evidence for conspecific recruitment for cooperative hunting in lionfish *Pterois miles*" is now accepted for publication in Royal Society Open Science.

on behalf of Prof Kevin Padian (Subject Editor)
openscience@royalsociety.org

Appendix A

Reviewer: 1

Comments to the Author(s)

I reviewed this manuscript upon its first submission to Biology Letters. Before I comment on this version of the manuscript, I first wanted to comment on some of the initial reviewer comments, and subsequent rejection of the manuscript from Biology Letters. One of the reviewers on the initial submission commented that the manuscript contained negative results and would be more appropriate for a specialized journal, despite the fact that this manuscript was a direct attempt to replicate the cooperative hunting behaviours reported in a previous Biology Letters paper that required a subsequent Expression of Concern. One of the major problems with science is that remarkable findings are hastily published with scant evidence, yet more robust studies are hastily rejected if they find less-glamorous results. This manuscript should have been published in Biology Letters to give it the same platform as the (problematic) lionfish cooperative hunting paper by Lonnstedt et al.

(<https://www.sciencemag.org/news/2019/09/can-you-spot-duplicates-critics-say-these-photos-lionfish-point-fraud>).

Following from the above, I strongly disagree with the authors removing the section about the Lonnstedt et al. issue in the introduction of the manuscript. This is very important information that underscores why the present study was conducted, so the information should be put back into the introduction.

#the suggested part is added lines 102-106

Aside from that major comment, I only have a few minor comments in addition to my last review.

Line 166 – fish were brought to the surface and placed immediately into holding tanks? Were they transported to a field lab first?

#Changed, lines 169-171

Line 383 – what is the transparent holding structure and why is that different from the Lonnstedt study?

explained in lines 384-388.

I commend the authors for supplying data and videos on dryad. Please make sure the data are the raw data and they are organised well, with a metadata tab to explain each column. Please also make sure that all videos are available if anybody wants them, not just ‘representative’ videos of particular behaviours.

#uploaded in dryad

https://datadryad.org/stash/share/dCZo_wZjl-WHbFH6Z480nvv1ll_xOkalecWW2g90iwQ

A thorough proofread is required to pick up several typos throughout.

proofed is attached

Reviewer: 2

Comments to the Author(s)

I enjoyed reading this interesting study which carried out some field and lab work which gave no support for cooperative hunting in *Pterois miles*. Lionfish have attracted attention as an invasive species and the potential for cooperative hunting behaviour in this context is intriguing. I don't see the negative results of this study as a problem but would like to recommend more careful phrasing of some sentences (including the title).

I don't think the authors can conclude that "lionfish *Pterois miles* do not recruit conspecifics for cooperative hunting" because they only tested for a limited number of possibilities. A simple solution would be to change the title to "No evidence for conspecific recruitment for cooperative hunting in lionfish *Pterois miles*". This change of title would probably be the most appropriate way of dealing with a negative result.

#changed

In the Abstract the authors state "...reinforcing the field results that this species in the Red sea does not depend on cooperation to catch fish". As far as I could tell, the authors did not observe any lionfish capturing prey in the field. They only recorded association patterns and fin flaring.

We reworded the line 30. cooperation to hunt fish

Statistical analysis

I am puzzled by Waser's gas model as an analysis tool. How can this possibly provide a prediction for biological processes? What if the lionfish like particular spots and like most animals don't use the entire available habitat to the same extent? I don't see how Waser's model accounts for micro-habitat preferences and how this can be distinguished from active association.

It is true that the model is not the best one. It is the only one that fits the quality of our data. Importantly, we did not calculate total time spent in association, which we think is more affected by micro-habitat preferences than the duration of single interactions (which only depend on swimming speed and the distance criterion). The model is qualitative in nature. The results are certainly not the most important ones but they fit the general impression that nothing is going on in our study population.

The information given on Lines 147-149 "The most basic measure was to quantify how much time lionfish spent in association with conspecifics and/or moray eels as described by Naumann and Wild [17]." is not very helpful because that is a single-page paper with no detailed description.

#clarified in lines 149-150

Minor points

Line 12, "Lionfish are highly successful piscivores"

Did you mean that they are highly piscivorous or that they are really particularly successful

compared to other species? If so, it would be interesting to know which species are used for comparison.

We reworded the statement. It is about abundance and being invasive in the Caribbean.

Lines 32-33 "Animals may benefit from grouping because of the transfer of information regarding the distribution of patchy food [1]."

There are many reasons why animals can benefit from grouping. Rephrase "One way in which animals may benefit from grouping is because of the transfer of information regarding the distribution of patchy food [1]" or cover more of the benefits of group-living.

#changed